# Continuous Co-Digestion of Agro-Industrial Mixtures in Laboratory Scale Expanded Granular Sludge Bed Reactors

Roberto Eloy Hernández Regalado [1,2,3,*], Jurek Häner [2,3], Daniel Baumkötter [2,3], Lukas Wettwer [2,3], Elmar Brügging [2,3] and Jens Tränckner [1]

1    Faculty of Agriculture and Environmental Sciences, University of Rostock, Justus-Von-Liebig-Weg 6, 18059 Rostock, Germany; jens.traenckner@uni-rostock.de
2    Faculty of Energy Building Services Environmental Engineering, Münster University of Applied Sciences, Stegerwaldstr. 39, 48565 Steinfurt, Germany; haener@fh-muenster.de (J.H.); baumkoetter@fh-muenster.de (D.B.); lukas.wettwer@fh-muenster.de (L.W.); bruegging@fh-muenster.de (E.B.)
3    Institute association for Resources, Energy and Infrastructure, Münster University of Applied Sciences, Stegerwaldstr. 39, 48565 Steinfurt, Germany
\*    Correspondence: roberto.hernandez@fh-muenster.de; Tel.: +49-159-0147-4299

**Abstract:** Anaerobic co-digestion often improves the yields and stability of single anaerobic digestion. However, finding the right substrate proportions within mixtures and corresponding optimal operating conditions using a particular reactor technology often presents a challenge. This research investigated the anaerobic digestion of three mixtures from the liquid fractions of piglet manure (PM), cow manure (CWM), starch wastewater (SWW), and sugar beet (SBT) using three 30 L expanded granular sludge bed (EGSB) reactors. The synergistic effects of two three-substrate mixtures (i.e., PM+CWM+SWW and PM+CWM+SBT) were studied using the PM+CWM mixture as a benchmark. These were used to detect the predicted synergistic interactions found in previous batch tests. The methane productivity of both three-substrate mixtures (~1.20 $L_{CH4}/L_{react}/d$) was 2× the productivity of the benchmark mixture (0.64 $L_{CH4}/L_{react}/d$). Furthermore, strong indications of the predicted synergistic effects were found in the three-substrate mixtures, which were also stable due to their appropriate carbon-to-nitrogen ratio values. Moreover, the lowest averaged solid to hydraulic retention times ratio calculated for samples obtained from the top of the reactors was > 1. This confirmed the superior biomass retention capacity of the studied EGSB reactors over typical reactors that have been used in agricultural biogas plants with a continuous stirred tank reactor.

**Keywords:** anaerobic co-digestion; synergistic effects; expanded granular sludge bed reactor; agro-industrial substrates

## 1. Introduction

Anaerobic digestion (AD) is an efficient and suitable method for the sustainable management of bio-wastes as well as the production of biofuel [1]. It is a biological degradation process whereby biomass is converted into a mixture of gases called biogas, which consist mainly of methane and carbon dioxide, by the action of a microorganism consortium in the absence of oxygen. It is typically divided into four main stages, including hydrolysis, acidogenesis, acetogenesis, and methanogenesis [2].

However, AD is a very complex and sensitive process, which involves diverse microorganism groups that require different environmental and operational conditions [1]. For example, biomass substrate digestibility and biogas production are significantly affected by the substrate composition and chemistry, such as the carbon-to-nitrogen ratio (C/N), mineral and volatile fatty acid composition, and pH [3]. These are also affected by the operational conditions, including the hydraulic retention time (HRT), substrate loading rate, reactor temperature, and so on.

High-efficiency energy production by AD has been commonly achieved through the use of a high organic substrate loading rate. However, the high loading rate affects the stability and efficiency of methane production due to the imbalance between acidification and methanation, which typically result in a significant accumulation of volatile fatty acids and a sharp decrease in pH, leading to system failure [4].

The most popular approaches to improving AD for biogas production include anaerobic co-digestion (AcoD), coupling with dark fermentation, microbial community bioaugmentation, reactor engineering, substrate pretreatments, and the use of enzymes as biocatalysts [1].

AcoD in particular involves the simultaneous digestion of two or more substrates. It has been shown to be a highly viable option for improving biogas production by alleviating the disadvantages of mono-digestion [4]. It also increases the economic feasibility of the process in existing AD plants by increasing methane yields [4]. The advantages of AcoD include the ample supply of macro and micronutrients, a balanced C/N, the dilution of reaction inhibitors, a superior buffering capacity, and the enhancement of biogas production [5]. Despite the advantages of AcoD, it also presents some drawbacks due to the inappropriate selection of co-substrates, co-substrate composition, and operating conditions. Consequently, a poorly researched co-digestion process may result in an instable process, bringing with it a significant reduction in methane production. It is therefore necessary to have a profound comprehension of the co-digestion mixture(s) employed at a lab and pilot plant in order to support full-scale design and operation decisions [3,6].

With 9632 operating and 9692 forecasted biogas plants in 2020 and 2021, respectively, Germany is the largest producer of biogas in Europe [7]. Its main feedstock for biogas production was initially energy crops. However, due to the current policy framework, Germany has shifted toward the use of alternative substrates such as crop residues, livestock waste, and catch crops [8,9]. Hence, agro-industrial wastes have gained importance due to their potential as raw materials for obtaining energy [10]. Their use could eventually reduce environmental liabilities and add value to already developed production chains. In 2018, 95% of Germany's mass-specific substrates came from animal excrements (48%) and renewable resources (47%) such as maize or grass silage [11]. AD has been carried out either as a single or co-digestion system. Production plants have been equipped with a gas-tight storage tank and a minimum of two digesters that are connected in series [12]. However, the optimization of HRTs is still required to reach high degradation values [11,12].

A large share of the studies on anaerobic co-digestion are concerned with the enhancement of biogas production while increasing methane content and shortening the retention time. Nevertheless, typical anaerobic digestion systems are not sufficiently efficient for today's demand [13]. The alternative may be a combination of modern reactors with enhanced biomass retention capacity and optimized digestion conditions (pH, temperature, HRT, among others) to obtain higher methane yields and productivity [8,13,14].

In the selection of a suitable bioreactor, the biomass retention capacity is an important consideration, because anaerobes grow slowly during the metabolic generation of butanol, ethanol, hydrogen, and methane [15]. This is particularly important in a bioreactor configuration that decouples HRT from solids retention times (SRT). These reactors, that are usually named high-rate reactors, were initially developed in the late 1970s with the introduction of the up-flow anaerobic sludge blanket (UASB) reactor [16].

The decoupling of HRT and SRT enables the maintenance of a significantly higher SRT/HRT ratio than in a continuous stirred tank reactor (CSTR) and prevents the washout of slow-growing anaerobes. Therefore, high-rate anaerobic systems are maintained at a sufficiently high biomass level inside the bioreactor [17]. In addition, environmental conditions are well preserved under optimal bioreactor performance parameters. The organic loading rates in these systems typically vary from 5 to 30 $kg_{COD}/(m^3 \cdot d)$, although higher rates have been reported [15,18].

The two main types of high-rate systems include suspended and attached growth. The expanded granular sludge bed (EGSB) reactor is a suspended high-rate system, which

has been used in industrial wastewater treatment. The implementation of EGSB reactors for biogas production has grown very fast in the last two decades [19]. Interestingly, EGSB reactors appeared as an improvement on UASB reactors, which allow high height–diameter relations for achieving high superficial velocities of >4 m/h EGSB per 1.5 m/h UASB [20,21]. EGSB reactor technology was developed to optimize internal mixing and solve problems which are typically found in the practical operation of UASB reactors, such as the occurrence of dead zones, preferential flows, and short circuits, among others. Consequently, EGSB reactors provide better substrate—biomass contacts within the treatment system by expanding and intensifying the sludge bed and hydraulic mixing, respectively [1,20].

This study aimed to assess the performance of the AcoD of three manure-based agro-industrial mixtures in three different EGSB reactors employed in a continuous operation mode. The AcoD of manure-based mixtures has acquired more relevance due to changes in the German Renewable Energy Sources Act (EEG), as a result of which only small liquid manure plants and waste digestion plants continue to benefit from the original remuneration system outlined in 2012 [11]. Furthermore, the German government encourages the use of natural fertilizers to reduce greenhouse gas emissions in the recycling of nitrogen [22]. The substrates were collected in the federal state of Nordrhein-Westfalen, Germany. In particular, the optimal compositions of two of the considered agro-industrial mixtures were determined using an approach initially designed by our group. The optimal composition of the third mixture was determined in Regalado et al. [23] using the approach of Regalado et al. [24]. The observed performance characteristics of the laboratory-scale reactors will form the basis for operation optimization and scale-up to pilot plant scale.

## 2. Materials and Methods

### 2.1. Mixtures and Inoculum Characterization

Mixture 1 was a combination of piglet manure (PM) and cow manure (CWM). It served as the benchmark to measure the change in performance by the addition of a third substrate, which was included in mixtures 2 and 3. Information on the mixtures and their inoculums is summarized in Table 1.

**Table 1.** Characterization of the mixtures and their inoculums.

| Reactor | Substrates | Dry Matter (wt.%) | Organic Dry Matter (wt.%) | Carbon-to-Nitrogen Ratio (%) |
|---|---|---|---|---|
| 1 | Pellets 1 | $7.79 \pm 0.16$ | $87.59 \pm 0.06$ | |
| | PM+CWM | $2.92 \pm 1.40$ | $60.08 \pm 9.32$ | 13.70 |
| 2 | Pellets 2 | $8.84 \pm 1.75$ | $89.47 \pm 1.45$ | |
| | PM+CWM+SWW | $1.76 \pm 0.94$ | $58.33 \pm 9.07$ | 16.32 |
| 3 | Pellets 3 | $7.93 \pm 0.29$ | $87.56 \pm 0.10$ | |
| | PM+CWM+SBT | $3.14 \pm 0.99$ | $64.55 \pm 8.52$ | 18.87 |

Piglet manure (PM); cow manure (CWM); sugar beet (SBT); starch wastewater (SWW).

### 2.2. Bioreactor Setup and Operation

Three EGSB reactors with a height–diameter ratio of 3 units were employed in a continuous operation mode. The reactors were inoculated with 20 L of mesophilic inoculum with a spherical shape and dark green color. The EGSB reactors were operated at six different HRTs for 15, 10, 7, 5, 3, and 1 day(d). The HRTs were automatically altered by changing the feeding time of the pump at a constant flow. The HRT was calculated by Equation (1) [19,25].

$$\mathrm{HRT} = \frac{V_R}{Q} \tag{1}$$

where the HRT, volume of the reactor ($V_R$), and influent volumetric flow are in day(s), m$^3$, and m$^3$/day units, respectively.

The recirculation pump was continuously working at an up-flow velocity of 5 m/h. Each reactor was connected to a 100 L tank that was kept under a nitrogen atmosphere and

temperature of 4 °C to prevent premature aerobic degradation. The scheme for a single reactor is shown in Figure 1.

All three reactors were operated under mesophilic conditions with temperatures between 37 and 40 °C and pH values close to 8 by regulating the feed of the reactor, which is mainly possible due to the buffer capacity of the manures [26,27]. The procedure for the settings and monitoring of the continuous operation was as described in reference [28]. The measured and set variables are summarized in Table 2. Other relevant values were calculated from registered variables such as the organic loading rate (OLR) ($kg_{COD}/m^3/d$), methane productivity (MPR) ($L_{CH4}/L_{reactor}/d$)), methane yield (MY) ($L_{CH4}/kg_{VS}$), removal efficiencies of chemical oxygen demand ($\eta_{COD}$) (%), and biological oxygen demand on the fifth day ($\eta_{BOD5}$) (%).

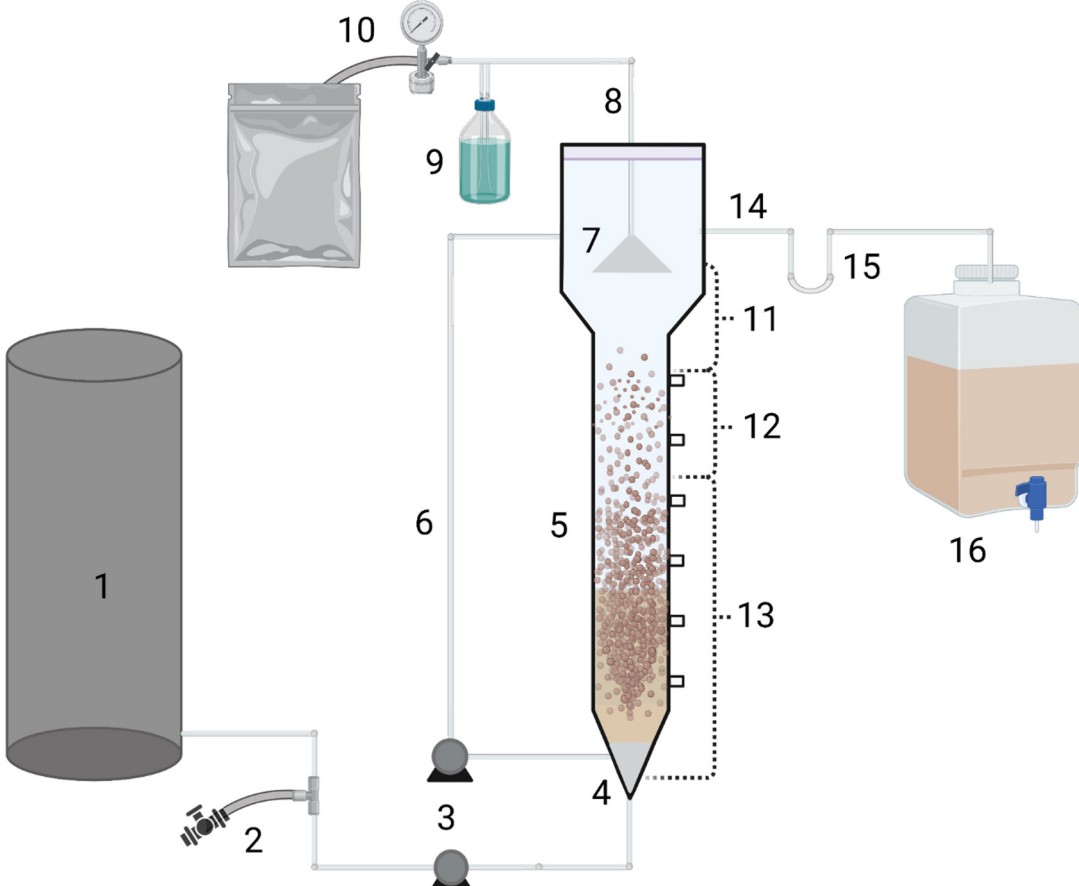

**Figure 1.** Schematic diagram of the reactor. The parts are (1) feed tank, (2) three-way sampling valve, (3) eccentric screw pumps, (4) mixer for influent and recirculation, (5) bioreactor, (6) recirculation, (7) bell separator, (8) biogas outlet, (9) foam trap, (10) gas flow-meter, (11) three-phase separator or settling zone, (12) transition zone, (13) digestion zone, (14) effluent, (15) siphon, and (16) digestate storage.

**Table 2.** Registered variables in the monitoring of reactor operations.

| Variable | Input | Inside the Reactor | Output |
|---|:---:|:---:|:---:|
| Temperature (°C) | | x | |
| Dry matter (%) | x | x | x |
| Organic dry matter (%) | x | x | x |
| C/N (%) | x | | |
| Chemical oxygen demand (mg$_{O2}$/L) | x | | x |
| Biochemical oxygen demand at 5th day (mg$_{O2}$/L) | x | | x |
| Loading rate per unit volume (kg$_{COD}$/m$^3$·d) | x | | |
| Hydraulic residence time (d) | x | | |
| pH value (-) | x | | |
| Gas composition in volume fractions (%, ppm) | | | x |
| Ratio of volatile organic acids to total inorganic carbon (-) | | x | |

x: measured in the coresponded position.

### 2.3. Data Cleaning and Analysis

Data cleaning was performed with the aim of obtaining a data set which did not contain obvious failures, start-up periods, or clear mistakes in an operation. The data cleaning was performed using the three main criteria as described below.

1.  The HRT is ≤30 day.
2.  The MY < biomethane potential of the mixture at HRT$_\infty$ (BMP$_\infty$), which was taken from Regalado et al. [24].
3.  The chemical oxygen demand removal is ≥0.

#### 2.3.1. Overview of Each Reactor's Operation

A comparison of the different operation points of a given reactor was made using the variables of MY, MPR, $\eta_{COD}$, and biological oxygen demand removal ($\eta_{BOD5}$). An analysis of the practical operation of each reactor was completed using this information and the complementary information on the mixtures involved. Box and scatter plots were employed to visualize each reactor's operation.

#### 2.3.2. Principal Component Analysis

PCA is an adaptive exploratory method which can be used on numerical data of various types. From a mathematical point of view, principal components are linear combinations of original variables, making them orthogonal to each other [29,30]. This method increases the interpretability of the data and at the same time minimizes information loss. For each reactor, a new data set was created using the average values of all the operation points for the above for all four response variables. The new datasets were used in a principal component analysis (PCA) to compare the reactors in terms of operation points as part of a multivariate analysis. Up to five components were acceptable and three components were desirable. The goal of the PCA was to rotate the data into an axis system where the greatest amount of variance was captured in a small number of dimensions [31].

The PCA involved the calculation of the eigenvectors and eigenvalues of a sample covariance or correlation matrix. Furthermore, the calculation of the principal components was carried out using a singular value decomposition (SVD) [32]. PCA was also employed for outlier detection due to its robustness [32,33].

### 2.4. SRT/HRT

A high-rate reactor such as an ESGB reactor can decouple HRT and SRT, thereby increasing the residence time of a biomass element within the reactor [15,34,35]. One of the main selection criteria for a reactor is a high SRT/HRT ratio, which prevents the washout of slow-growing methanogens [15].

The sludge age (SRT) in d is given by Equation (2).

$$\text{SRT} = \frac{\text{Mass of sludge in reactor}}{\text{Mass of sludge wasted per day}} \ (\text{d}) \tag{2}$$

If a steady-state condition was assumed, Equation (2) can be written as Equation (3).

$$\text{SRT} = \frac{x_i \cdot V_R}{Q_{eff} \cdot X_{eff}} \ (\text{d}) \tag{3}$$

where $x_i$, $V_R$, $Q_{eff}$, and $X_{eff}$ are the viable biomass concentration inside the reactor ($kg_{VS}/kg_{FM}$), volume of the bioreactor ($m^3$), effluent flow out (L/d) of the reactor, and viable biomass concentration in the effluent ($kg_{VS}/kg_{FM}$), respectively. Since the input and output flows were equal (steady-state condition), Equation (3) was transformed to (4).

$$\frac{\text{SRT}}{\text{HRT}} = \frac{x_i}{X_{eff}} \ (\text{d}) \tag{4}$$

The ratio SRT/HRT values were calculated for all three reactors using a biomass, which was sampled from the top of each reactor where biomass concentration was lowest. The ideal SRT/HRT ratio should be >3 [15,17,19].

*2.5. Characterization of Synergistic Effects*

The synergistic effects of the three-substrate mixtures (PM+CWM+SWW and PM+CWM+SBT) were compared using the two-substrate mixture (PM+CWM) as a benchmark. Since it was not possible to operate a single digestion of each substrate, the hypothesis employed used Equation (5) to validate the synergistic effects.

$$\left( \frac{MY_{MAX\_TM}}{MY_{MAX\_DM}} \right)_{continuous} \cong \left( \frac{MY_{MAX\_TM}}{MY_{MAX\_DM}} \right)_{batch} \tag{5}$$

where $MY_{MAX\_TM}$ and $MY_{MAX\_DM}$ are the maximum MY of the three- and two-substrate mixtures, respectively.

Since the calculated ratios were based on the yield of the two-substrate benchmark mixture, the ratio for the PM+CWM mixture was equal to 1. As for the batch data results from Regalado et al. [24] and some complementary unpublished data, the $MY_{MAX}$ value corresponding to a maximum value of each operating point was used. In addition, a comparison between the $MY_{MAX}$ in the continuous and batch operation processes was made for each mixture.

*2.6. Characterization of Hydraulic Behaviors*

The hydrodynamics of the anaerobic reactor was studied because they significantly influence the rates of biological reactions. They particularly affect the rates of mass transfer and the distribution of reactions along a reactor, both of which determine a reactor's overall performance [20,36]. The amount of mixing in a reactor also determines the performance of a reactor; therefore, to describe the real behavior of a reactor, the influence of mixing on the mass balance equation must be specified correctly [37]. In this study, the hydrodynamics were characterized by the non-dimensional numbers given by Peclet and Reynolds.

The mixing intensity of the fluid within a reactor is well described by the axial Peclet number ($Pe_{axial}$) (see Equation (6)).

$$Pe_{axial} = \frac{V_{up} \cdot H}{D_A} \tag{6}$$

where $V_{up}$, $H$, and $D_A$ are the up-flow velocity (m/h), bioreactor height (m), and axial dispersion coefficient ($m^2/h$), respectively. When $D_A \rightarrow \infty$, the value of $Pe_{axial}$ becomes 0 since $Pe_{axial}$ is an inverse function of $D_A$. Consequently, the system will operate as

a plug-flow reactor since there is no mixing in the axial direction. On the other hand, when $D_A \rightarrow 0$, the system will behave as a complete mixture reactor [19]. Various transfer functions have been proposed to estimate the dispersion from either the Reynolds number or a flow velocity [38–42]. Here, we used an approach described by Equations (7) and (8), which assessed $D_A$ as a function of flow distance

$$D_A = 1.03 \cdot V_{up}{}^{1.11} \cdot 0.009^{n_j} \tag{7}$$

$$n_j = \frac{z}{H} \tag{8}$$

where $n_j$, z, and H are the values of the normalized height, axial position (m), and height (m) of the bioreactor, respectively.

The amount of turbulence is characterized by the Reynolds number (Re) and is given by Equation (9) [19,43]. Where $V_{up}$, d, $\mu_w$, $\rho_w$, and $\upsilon_w$ are the up-flow velocity (m/h), bioreactor diameter (m), dynamic viscosity (Pa·s), density (kg/m$^3$), and kinematic viscosity (m$^2$/s), respectively, Reynolds describes a relationship between inertial to viscous forces [42]. Equation 9 is the most widely used; however, it exits variations of the Reynolds number around noncircular conduits, packed beds, and mixing impellers.

$$Re = \frac{V_{up} \cdot d}{\upsilon_w} = \frac{\rho_w \cdot V_{up} \cdot d}{\mu_w} \tag{9}$$

Turbulence, meanwhile, is the rotational and three-dimensional chaotic movement in all directions of flowing elements, where the resulting net flow is unidirectional. The rapid mixing associated with turbulence enhances the momentum, heat, and mass transfer processes. The intervals of Reynolds include Re < 2300, 2300 < Re < 4000, and Re > 4000, which correspond to laminar, transient, and turbulent regimen, respectively. However, a typical turbulent regimen truly manifests itself from values of Re > 10,000.

### 2.7. Modeling of Reactors
Stover–Kincannon Model

The MPRs of the reactors were modeled using a variation of the Stover–Kincannon model for an anaerobic filter reactor (Equation (10)), which was proposed and implemented by Yu et al., Verma et al., Jafarzadeh et al. [44–46].

$$MPR = \frac{MPR_{max} \cdot OLR}{M_B + OLR} \tag{10}$$

where MPR, $MPR_{max}$, and $M_B$ are the methane productivity ($L_{CH4}/L_{react}/d$), maximum MPR ($L_{CH4}/L_{react}/d$), and constant ($kg_{COD}/m^3 \cdot d$), respectively. OLR is the organic loading rate ($kg_{COD}/m^3 \cdot d$). A non-linear regression procedure was employed using the calculated clean averaged data of all reactors. To identify similarities and differences in the kinetic behavior of all possible combinations, the averaged data of reactors 1, 2, and 3 were arranged to have a total of seven datasets. The goodness of fit was measured by a root-mean-square error (RMSE). For the most meaningful dataset(s), a simple regression analysis was performed for MY and $\eta_{COD}$ removal using OLR as the independent variable. The goodness of the fit was compared by the $R^2$ value, simplicity, and Durbin–Watson coefficients (D–W) to determine the most significant dependency. $R^2$ values of <0.7 were automatically dismissed and those >0.8 were identified as desirable.

### 2.8. Reactor's Optimization

Once the significant models were identified, their dependencies were plotted with OLR to perform a graphical optimization. In a graph, the ordinates represented the values of the individual variable divided by their maximum measured value ($V_i/V_{Max}$), which was expressed in %. Thus, the ordinates represented values between 0 and 100% for each plotted variable.

## 3. Results

### 3.1. Reactors' Operation Overview

Data cleaning was performed according to the set criteria in Section 2.3. The two-substrate mixture of PM+CWM (pellets 1) had a $BMP_\infty$ of 342.83 $L_{CH4}/kg_{VS}$. Meanwhile, the $BMP_\infty$ values for the three-substrate mixtures of pellets 2 and 3 were 534.21 and 530.28 $L_{CH4}/kg_{VS}$, respectively [24,47]. After data cleaning, the resulting data sets have sizes of 162, 181, and 203 instances for reactors 1, 2, and 3 respectively. To accurately characterize the performance of a reactor, four employed output variables, including MPR ($L_{CH4}/L_{reactor}/d$), MY ($L_{CH4}/kg_{VS}$), $\eta_{COD}$, and $\eta_{BOD5}$ (%), were used [28,48]. The overview of reactor 1 is shown in Figure 2. The MPR suffered a sudden drop at the operating point of 5 d. This interrupted the upward trend that was observed from HRT of 15 to 7 d. From 3 to 1 d, a sustained increase in the MPR was observed, with a value that was almost 2× the second-highest average value observed at 7 d. This suggested that a punctual failure occurred at 5 d, which was not due to reaching an operational HRT limit. MY had the highest mean value of 272 $L_{CH4}/kg_{VS}$ at 15 d. After 15 d, the values significantly decreased and a similar sudden drop in MPR was observed at a HRT of 5 d. However, MY did not experience a significant recovery after the inhibition, unlike MPR. Thus, by considering both variables of MPR and MY simultaneously, the operation can be divided into two main stages: before and after inhibition. There is a noteworthy difference between these two stages. The former reached considerably higher yields than the latter; however, similar values of MPR were found in both stages.

The removal efficiencies in these two stages were not as evident. $BOD_5$ removal efficiency values noticeably dropped at 10 and 1 d. The low $BOD_5$ removal values may explain the drop in MY at 10 and 1 d in the previous operation point. This was most likely due to low reaction completions [49]. A minimum average value for the chemical oxygen demand (COD) removal efficiency was observed at 5 d. However, the trend followed by the average $\eta_{COD}$ values had a smaller variation compared to both MPR and MY. Furthermore, both $BOD_5$ and removal efficiencies behaved differently since the calculated $BOD_5/COD$ ratios were fluctuating.

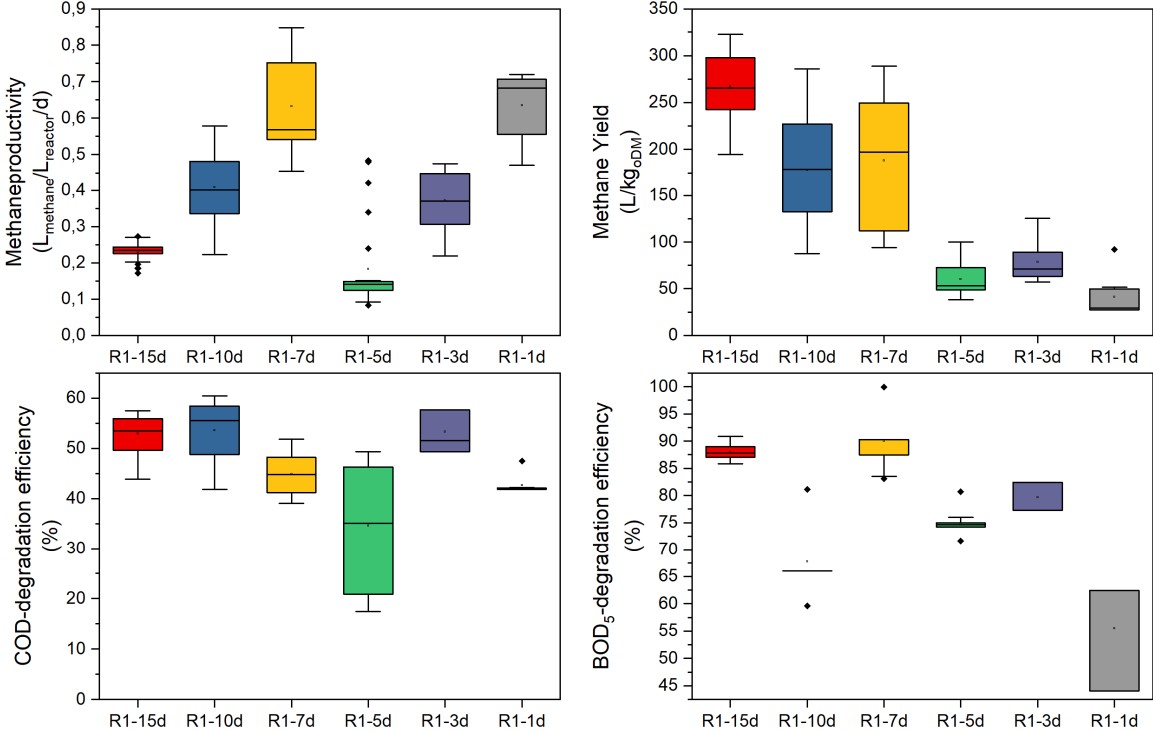

**Figure 2.** Summary of the main response variables for reactor 1. ●: averaga values; ✦: outliers.

To identify the potential causes of inhibition, the control variables of OLR and VOA/TIC were employed. The results are summarized in Figure 3. The high OLR values observed were not an indicative cause of inhibition. In particular, the OLR value was not very high at 5 d. Moreover, the system did not run at very high OLR values despite a constant decrease in the HRT values. Therefore, the COD values in the inflow suffered sizable fluctuations, as shown in Figure 4.

OLR fluctuations were not strongly correlated with the VOA/TIC, which implies that a failure was not due to a system overload caused by overfeeding. Instead, they seemed to be more connected to the quality of the feed (Figure 4). However, the VOA/TIC results showed that an acid accumulation occurred at 10 d. In these results, the values decreased sharply and approached zero at the operating point of 5 d. Hence, two possibilities were weighted, such as the non-failure of a system due to OAs accumulation and a system delayed response due to VOA accumulation at 10 d. In the former, there was no sign of strong inhibition before 5 d and a lack of VOAs in the system. This were supported by the close to zero VOA/TIC values. In the latter, the system had a delayed response to VOA accumulation at 10 d, which seemed more unlikely, given how large the delay had to be. In addition, the MPR values increased from 10 to 7 d, while the MY values were practically the same. Therefore, either the acetogenesis was the limiting-rate step or the quality of the feed was very low.

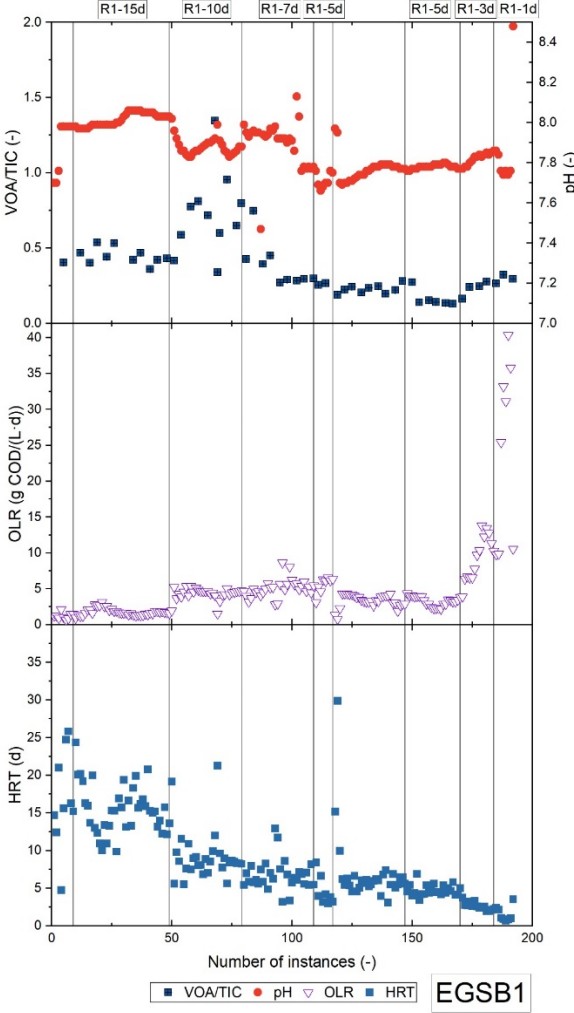

**Figure 3.** Volatile organic acids to total inorganic carbon ratio (VOA/TIC), organic loading rate (OLR), and hydraulic retention time (HRT) in reactor 1.

$COD_{in}$ and $VOA_{s\_in}$ values were determined for the feed to investigate whether the quality of the feed was responsible for the inhibition. The results are summarized in Figure 4. All the targeted acids were found except for valeric and caproic acids. A rapid decline in the total acetic acid concentration and equivalent occurred after 10 d. Likewise, the concentration of the acids was almost zero at 5 d. This behavior was consistent with the inhibition observed at the operating point of 5 d. Thus, the occurrence of a failure due to the lack of VOAs in the feed, which caused very low VOA/TIC values, was accepted.

The observed $COD_{in}$ fluctuations explained the behavior of the OLR with the gradual reduction of HRTs. It was expected that intensive variables, such as $COD_{in}$, VOA concentrations, and the $BOD_5/COD$ ratio, would show stable behavior. However, these variables fluctuated due to the lack of proper mixing in the feeding tank. Continuous mixing was not done during operations, although the mixtures were vigorously mixed in the tank during preparation. This induced the settling of particulates and instability of the feed, which caused the first excess of VOAs observed at 10 d. The concentration of VOAs was approximately zero at 5 d, which suggested the existence of a substrate limitation on the system. This limitation substantially influenced the operation since the tank had to be refilled several times, causing variations in the preparation. Therefore, heterogeneities in the composition of substrate mixtures during a year of operation are expected due to seasonal behavior [50].

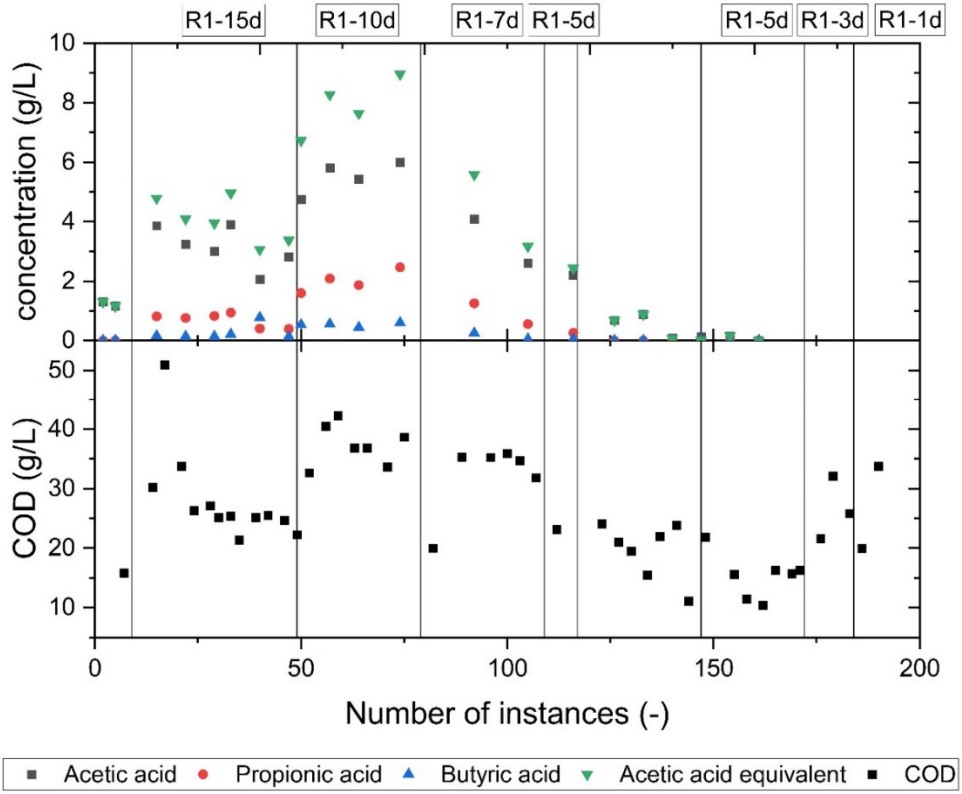

**Figure 4.** Volatile fatty acids and chemical oxygen demand (COD) in the feed of reactor 1.

Similar trends to reactor 1 were observed for reactors 2 and 3. For reactor 2 in particular, the inhibition was the least abrupt due to a lower dry matter (DM) content (i.e., almost 80% of SWW on a fresh matter basis), which reduced the effect of the seasonal behavior observed in the manures (Table 1). A summary of the averaged behavior for a specific operation point for each reactor is found in Table 3, which uses a three-color scale by column. The colors were ordered red, yellow, and green to show the increase from lower to higher values. The intensity of each color was determined by its proximity to the lowest, middle, or highest value.

**Table 3.** The averaged values by operation point for each reactor.

| Operating Points | Mean Values COD Removal Efficiency (%) | | | Mean Values of Methane Yield ($L_{CH4}/kg_{VS}$) | | | Mean Values of Methane Productivity ($L_{CH4}/L_{react}/d$) | | | Mean Values BOD$_5$ Removal Efficiency (%) | | |
|---|---|---|---|---|---|---|---|---|---|---|---|---|
| | R1 | R2 | R3 | R1 | R2 | R3 | R1 | R2 | R3 | R1 | R2 | R3 |
| Operating Point 1 (15 d) | 52.7 | 49 | 59.5 | 265.7 | 295.6 | 217.6 | 0.23 | 0.15 | 0.29 | 88.13 | 84.98 | 76.84 |
| Operating Point 2 (10 d) | 53.6 | 58.6 | 76.2 | 180.2 | 261.6 | 382.5 | 0.41 | 0.21 | 0.83 | 68.77 | 83.64 | 85.34 |
| Operating Point 3 (7 d) | 45.3 | 82.5 | 63.7 | 189.2 | 395.8 | 317 | 0.64 | 0.46 | 0.96 | 90.16 | 95.22 | 97.69 |
| Operating Point 4 (5 d) | 34.2 | 73.9 | 53.8 | 61.1 | 158 | 158.8 | 0.19 | 0.42 | 0.6 | 73.03 | 90.93 | 81.23 |
| Operating Point 5 (3 d) | 53.2 | 77.3 | 55.7 | 78.1 | 200.3 | 193.9 | 0.34 | 0.91 | 0.79 | 78.21 | 91.09 | 77.47 |
| Operating Point 6 (1 d) | 42.6 | 64.1 | 43.7 | 41.4 | 77.6 | 89.4 | 0.64 | 1.2 | 1.21 | 84.56 | 92.77 | 84.56 |

The colors are ordered red, yellow, and green from lower to higher values. R is the reactor.

The maximum value of $\eta_{COD}$ removal efficiency observed in reactor 1 was much smaller than in reactors 2 and 3. The same trend was observed with MY. The COD removal efficiency has been interpreted as a degree of reaction completion [19,41,49]. Thus, a strong relationship between COD removal and reaction completion was expected. Meanwhile, a similar relationship should have been found between MY and $\eta_{BOD5}$; however, due to fluctuations in the $BOD_5$/COD, no clear visual correlation could be established. Furthermore, the drop in the MYs of all three reactors at HRT 5 d was linked with the lack of organic dry matter (ODM) or VOA in the feed, as previously shown in Figure 4.

The MYs in reactors 2 and 3 were also significantly higher than those in reactor 1. The maximum MY of reactors 2 and 3 were similar at 7 and 10 d, respectively. The improvement observed thanks to the addition of a third substrate to the PM+CWM mixture was probably related to a higher C/N ratio. The C/N ratio balance in feedstocks was significant for the stable operation of AD. Substrates with high C/N ratios have a poor buffering capacity; therefore, nitrogen will be consumed rapidly by methanogens to meet their protein requirements. This results in low methane production and produces excess VOAs during fermentation. In typical feedstocks with a low C/N ratio, nitrogen has been found to accumulate in the form of ammonia, which inhibits the methanogens and prevents methane production [4,51].

The lowest recommended limit for C/N is 20 [11]; thus, a value of 15 was sufficient for our purpose [13]. The values of C/N in Table 1 are between 15 and 20 for the three-substrate mixtures. Meanwhile, the value was <15 for the two-substrate mixture. This supported our finding that the three-substrate mixtures were more stable and produced more methane. Nonetheless, Lissens et al. [52] have affirmed that substrates with a C/N ratio <10 can support a stable process; however, they require a multistage system to avoid reactor overloading.

The maximum values observed for the COD and $BOD_5$ removal efficiencies, as well as the MY for reactors 2 and 3, were at HRTs of either 10 or 7 d. The same operation interval was observed by Cruz-Salomón et al. [19] and regarded as the optimal operation interval for EGSB reactors.

All three reactors reached their maximum MPR at 1 d. The observed stable operation of the EGSB reactors at HRTs of 3 and 1 d presented some novelty in our operation with the agricultural substrates. Castrillón Cano et al. [53] were able to operate reactors at HRTs of as low as 8 h; however, they only used a 3.4 L effective volume to perform their residence time distribution (RTD) experiments with water in the presence and absence of biomass. In another study, Dereli [54] effectively operated a full-scale EGSB of 1200 $m^3$ at an average HRT of 7 d for the treatment and digestion of confectionery industry wastewater. Meanwhile, Cruz-Salomón et al. [25] performed continuous tests with a 3.3 L EGSB reactor with a HRT of between 3 and 9 d for the treatment of coffee processing wastewater. In addition, Rico et al. [55] operated a UASB reactor with an external settler and effluent recycling for alkalinity supplementation for the co-digestion of cheese whey and the liquid fraction of dairy manure. Under a constant HRT of 2.2 d, their system demonstrated a stable operation with up to 75% cheese whey fraction in the feed. This was with an applied OLR of 19 $kg_{COD}$ $m^{-3}$ $d^{-1}$, obtaining a $\eta_{COD}$ and MPR of 94.7% and 6.4 $m^3_{CH4}$ $m^{-3}$ $d^{-1}$, respectively. They observed critical biomass washout when the cheese whey fraction in the feed was 85% for a HRT of 2.2 d. Operation at a constant cheese whey fraction of 60% in the feed mixture enabled a stable operation under an OLR and HRT of 28.7 $kg_{COD}$ $m^{-3}$ $d^{-1}$ and 1.3 d, respectively. In addition, the $\eta_{COD}$ and MPR values were 95.1% and 9.5 $m^3_{CH4}$ $m^{-3}$ $d^{-1}$, respectively. Therefore, we conclude that there is a novelty in our successfully operated EGSB reactors for AD from the agricultural substrates in mixtures 1 and 3 at small HRTs. Notably, Rico et al. [55] suggested that a manipulation of the mixture proportion at a constant HRT can also lead to improvements in terms of both stability and MPR.

### 3.2. PCA

PCA was conducted using the data shown in Table 3. The results are shown in Figure 5. The score of Figure 5a shows the distribution of the data in the reactor number combining the shapes and colors shown in the figure legend. The HRT is shown above each point. The axes of the graph were created by a linear combination of the variables. This is represented in Figure 5c with MY and COD removal being the most influential variables in the x and y-axis. respectively. Meanwhile, Figure 5b shows the combination of Figure 5a,c. The red and blue points are the variable and data points, respectively. The x-axis was the most significant since it explained most of the variability of the data (e.g., up to 99%). PCA has been used for reducing the dimensionality of large datasets [29,30,56]. However, since the dataset employed was small, PCA was used for descriptive purposes only.

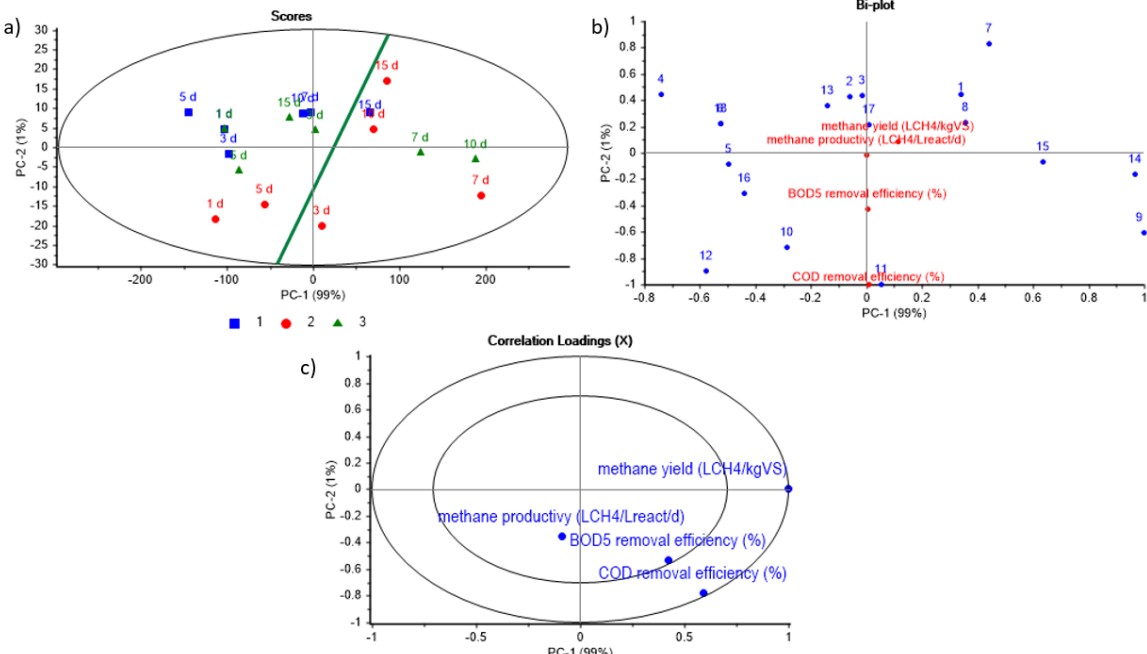

**Figure 5.** PCA overview of the averaged data. (**a**) Score plot, (**b**) bi-plot, and (**c**) correlation loading.

The MPR was not able to sufficiently describe the variability of the data since the most influential variables were found between the two external ellipses in Figure 5c. MY was the most important and efficient according to the PCA results within the low-right quarter of the ellipse in Figure 5a. The green diagonal line in both Figure 5a,b represent the difference between the efficient and non-efficient operations. Hence, the best points were 7, 10, and 7 d for reactors 1, 2, and 3, respectively. These points reached their high values simultaneously in all four variables (Table 3). The operation of reactor 1 reached a comparable efficiency to reactors 2 and 3 at a HRT of 15 d only. Nonetheless, a more efficient operation at lower HRTs was possible with the three-substrate mixtures.

Most of the operating points for HRTs at <7 d were considered inefficient, except for reactor 2 with a HRT of 3 d. Therefore, the HRTs at <7 d were generally feasible; however, they are not recommended due to their low efficiencies.

### 3.3. SRT/HRT

EGSBs can potentially reach much lower HRTs than CSTR reactors, which are typically used in agricultural biogas plants. In these reactors, the HRTs are equal to SRTs. However, an ESGB reactor can decouple the retention times by increasing the residence time of the biomass within the reactor [15,34,35]. The SRT/HRT values were calculated by equation 4 for each reactor using a biomass sample from the top of a reactor, where the biomass

concentration was lowest. A steady-state condition was assumed in the calculations and the results are shown in Table 4.

**Table 4.** SRT/HRT results for the three reactors (R1, R2, and R3).

| Operating Points | Solid Retention Time to Hydraulic Retention Time Ratio | | |
|---|---|---|---|
| | R1 | R2 | R3 |
| Start-up (15 d) | 1.10 | 1.29 | 1.24 |
| Operating Point 1 (15 d) | 1.72 | 1.41 | 1.59 |
| Operating Point 1 (15 d) | 0.82 | 0.83 | 1.16 |
| Operating Point 4 (5 d) | 1.22 | 1.33 | 1.15 |
| Reactor recovery (5 d) | 1.30 | 1.68 | 1.18 |
| Operating Point 4 (5 d) | 1.20 | 1.42 | 0.99 |
| Operating Point 5 (3 d) | 1.22 | 1.28 | 1.11 |
| Average value | 1.23 | 1.32 | 1.20 |

The red and green highlighted numbers were the lowest and highest values in the columns, respectively.

The ideal SRT/HRT ratio should be >3 [17]; however, this was far from being accomplished by sampling from the top of a reactor. In all cases, the averaged SRT/HRT was >1. Nevertheless, in three instances (one from each reactor), SRT/HTR ratios smaller than 1 were calculated. This demonstrates that even by sampling from where the biomass concentration is lowest, an average EGSB reactor can retain a better biomass than a typical CSTR. Minimal washout was observed in all three reactors. Unfortunately, data was not collected between the HRTs of 10 and 7 d.

### 3.4. Characterization of Synergistic Effects

To study the possible synergistic effects suggested by the interactions identified as acute effects in Regalado et al. [24], the ratios between the maximum MYs in the continuous operation and batch validation tests were compared using equation 5. The results are shown in Table 5.

**Table 5.** Methane yields and ratios based on piglet and cow manure yield in the batch and continuous tests.

| Mixture | Maximum Methane Yield in Continuous Tests ($L_{CH4}/kg_{VS}$) | Methane Yield Ratios in Continuous Tests | Methane Yield Predicted by the Model in Batch Tests ($L_{CH4}/kg_{VS}$) | Methane Yield Ratios in Batch Tests | Continuous to Batch Methane Yield Ratio |
|---|---|---|---|---|---|
| PM+CWM | 265.70 | 1.00 | 342.83 | 1.00 | 0.78 |
| PM+CWM+SWW | 395.80 | 1.49 | 513.07 | 1.50 | 0.77 |
| PM+CWM+SBT | 382.50 | 1.44 | 530.76 | 1.55 | 0.72 |

The MY ratio of the mixture with SWW was almost the same in both scales. The relative difference in the ratios obtained for the mixture with SBT was 7.01%. This confirmed the acute effects of adding a third substrate to the two-substrate mixture. The third substrate provided the same boost in the continuous stage and batch tests.

Also, the methane yield ratio obtained during the transfer from batch tests to the continuous stage was between 0.72 and 0.78. It was expected that the obtained MY from the continuous stage would be smaller than the ultimate biomethane potential from the batch test described by Weinrich and Nelles [11]. Similar intervals in the continuous stage to batch tests methane yields ratio have been identified in the literature. For example, Mahnert et al. [57] obtained ratios from 0.73 to 0.8 from the use of different grass species. Obiukwu and Nwafor [58] reached a ratio of 0.81 from the use of grape pomace. Meanwhile, Chowdhury and Fulford [59] used the mesophilic digestion of cattle dung in both batch and semi-continuous digestion with four reactors and six semi-continuous reactors, respectively.

Their results showed higher rates in the semi-continuous operation; however, biogas yields were lower compared to the batch test. Their batch tests reached 67% COD efficiencies, which was lower than the results in this study. In addition, Holliger et al. [60] suggested that BMPs can be used to estimate biogas production at full scale; however, the BMP value should be multiplied by a factor of 0.8–0.9 to avoid overestimation.

### 3.5. Hydraulic Analysis

The results for both hydraulic parameters of the reactors are shown in Table 6.

**Table 6.** Hydraulic parameters of the reactors.

| Parameter | Influent | Reactor Tube | Separation Zone |
|---|---|---|---|
| Re | 295.42 | 295.42 | 295.42 |
| $Pe_{axial}$ | 1.10 | 9.33 | 38.87 |

The $Pe_{axial}$ results showed values that were very close to 0, even in the separation zone. This indicated a flow pattern that was close to a completely mixed system [61]. The value of the Reynolds was also very similar to the one obtained by Brito and Melo [36]. These authors fitted an EGSB reactor to a CSTR with a characteristic coefficient of determination of 0.92. The inclusion of a short circuit increased the coefficient of determination to 0.95. Therefore, a CSTR model for simplicity was accepted and successfully used in their mass balance equations. Similarly, López and Borzacconi [62] assumed a CSTR behavior based on a high recirculation ratio and expansion of a bed. The combination of these two effects resulted in the significant mixing of the liquid and solid phases, as well as uniform gas production. However, their mass balance equation for the biomass included a washout effect, which was attributed to the high up-flow velocity at which the reactor was operated.

Nevertheless, the relative increase of the Peclet's value from one zone to another was considered significant. Consequently, due to the different behaviors of the zones in the reactor, the zones can be modeled as different reactors in series as described by Gleyce et al. [20]. Gleyce et al. [20] divided a reactor into two major zones, i.e., the separator and the reactor tubes. The reactor could be modeled either as two plug-flow reactors in series or five CSTRs with three separators and two tubes with coefficients of determination of 0.94 and 0.95, respectively.

### 3.6. Modeling of Reactors

The Stover–Kincannon model was used to fit the five different combinations of data from the reactors. The averaged values were employed and the size of a combination was up to 18 points. The results are summarized in Table 7. The fit for R2 was very good and the best among all datasets. The datasets for R1 and R3 had the worst fit, which suggested that the largest difference in the kinetic behavior among all possible combinations existed in these reactors. The RMSE values for R1 and R2 were also rather large, which meant that there were significant differences in the behaviors of these reactors. The goodness of fit in R1 and R3 were equal; however, R3 could produce a maximum amount of methane that was more than double the daily amount of methane from R1. R3 was also able to handle a larger OLR, which was related to the intrinsic properties of the substrate mixtures. In addition, the model predicts that R2 was far from reaching its maximum in terms of the production that can be handled by its largest OLR.

**Table 7.** Fit analysis using the Stover–Kincannon model.

| Data Set | RMSE ($L_{CH4}/L_{react}$/d)(-) | $M_{max}$ ($L_{CH4}/L_{react}$/d) | $M_B$ (gCOD/L/d) |
| --- | --- | --- | --- |
| (R1, R2, R3) | 0.214 | 1.29 | 6.08 |
| (R1, R2) | 0.188 | 1.25 | 8.21 |
| (R1, R3) | 0.245 | 1.07 | 4.20 |
| (R2, R3) | 0.118 | 1.48 | 5.07 |
| (R1) | 0.132 | 0.68 | 3.59 |
| (R2) | 0.031 | 1.76 | 9.99 |
| (R3) | 0.132 | 1.52 | 5.12 |

R: reactor; RMSE: root-mean-square error.

The datasets for R2 and R3 were of special interest. Since the mixtures in these reactors showed synergistic effects and they seemed to behave similarly, we examined if the mixtures followed similar kinetics. We found that the difference among them was moderate (R2, R3); however, the differences between R1 and R2 (R1, R2) or R1 and R3 (R1, R3) were larger. The fits and measured data are shown in Figure 6. By following both estimated models (red and violet lines), it was noticed that significant differences existed at low OLRs, with these being much smaller at higher OLRs. The fluctuations in R3 between 4.5 and 6.5 gCOD/L were most likely the main cause of the misfit, which was observed in the green but not in the red line. The performance of R3 (green line) was closely related to mixture preparation and the degree of mixing in the feeding tank since mixture 3 had the highest DM content, contrary to the smooth behavior of R2 (light blue line).

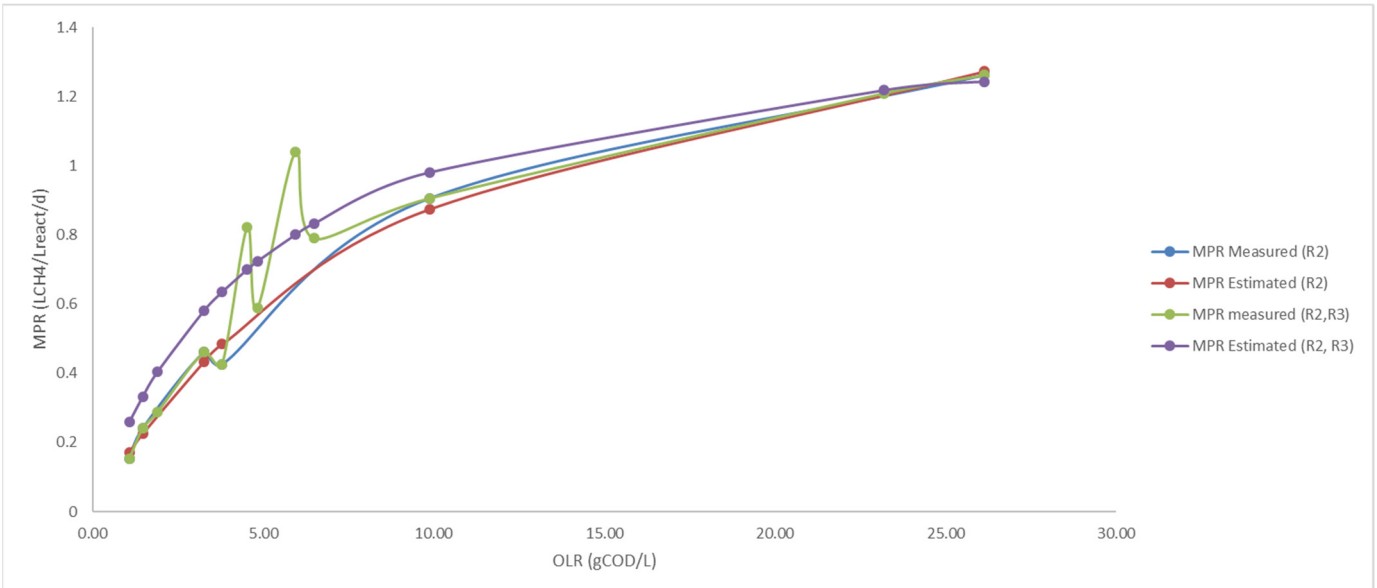

**Figure 6.** Stover–Kincannon model fitting for the datasets of R1, R2, and R3.

Therefore, we decided to work on both mixtures individually, given that both models were not able to converge in most of the working intervals. Since working at low HRTs usually reduces the MY [44], we took into account the other response variables in order to establish an optimal operational OLR. Hence, empirical models of MY and $\eta_{COD}$ versus OLR were also fitted. The fits for reactors 2 and 3 are shown in Figure 7a,b, respectively.

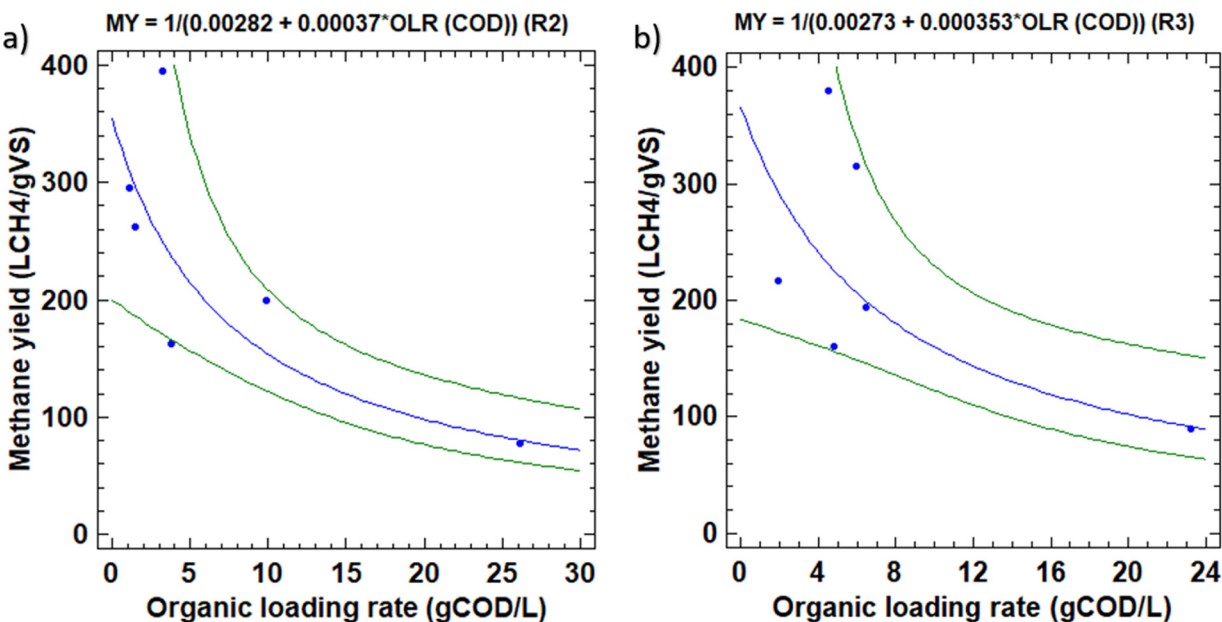

**Figure 7.** Comparison of the Stover–Kincannon model against measured data for EGSB reactors (R2 and R3). Blue and green lines corresponded to the model and confidence limits of the model at 95%, respectively.

No significant fit was found for $\eta_{COD}$, as per the pre-established criterion set for $R^2$. The $R^2$ values for R2 and R3 were 0.649 and 0.635, respectively. Therefore, no strong dependency on OLR existed. The fits should be described by more complex models that take into account mass transfer relationships. The MY was satisfactorily described by its inverse relationship with OLR for R2. The $R^2$ was 0.881 with a D-W of 3.6 (*p*-value = 0.990). While the fit for R3 was smaller, the $R^2$ of 0.782 with D-W of 2.22 (*p*-value = 0.4709) was still significant. The inverse relationship between MY and OLR has been described by several authors [44–46]. Since both the p-values above were greater than 0.05, there was no indication of serial autocorrelation in the residuals at the 95.0% confidence level. The fitting for both reactors is shown in Figure 7. All the points were contained or at least very close to the confidence limits of the prediction lines (green lines). Therefore, with all the above information combined, the models were considered acceptable.

### 3.7. Optimization of a Reactor

The two equation-systems developed for R2 and R3 combined the Stover–Kincannon model and reciprocal model for MY. Hence, the optimal OLR to simultaneously optimize MPR and MY for R2 and R3 are described by Equations [(11) and (12)] and [(13) and (14)], respectively.

$$\text{MPR} = \frac{1.76 \cdot \text{OLR}}{9.99 + \text{OLR}} \tag{11}$$

$$\text{MY} = \frac{1}{0.00282 + 0.00037 * \text{OLR}} \tag{12}$$

$$\text{MPR} = \frac{1.52 \cdot \text{OLR}}{5.12 + \text{OLR}} \tag{13}$$

$$\text{MY} = \frac{1}{0.00273 + 0.000353 * \text{OLR}} \tag{14}$$

The graphical optimizations are shown in Figure 8. The ordinates represent the % of the $\text{MPR}_{max}$ or the $\text{MY}_{max}$ measured. The call-out represents the point where both functions met each other. R2 can handle a higher OLR; however, the yields were less from

both functions than R3. Nevertheless, both reactors have similar working intervals, which provided reasonable yields from 3 to 5 gCOD/L for both response variables.

Using the averaged value of the COD in the feed, the working intervals were between 4 and 7 d and 6.5 and 11 d for mixtures 2 and 3, respectively. The upper value for mixture 3 was slightly above 10 d (HRT), the interval for agro-industrial wastewaters suggested by Cruz-Salomón et al. [25]. Meanwhile, mixture 2 had a working interval that was slightly lower than the selected interval. This was attributed to the lower COD content in the mixture.

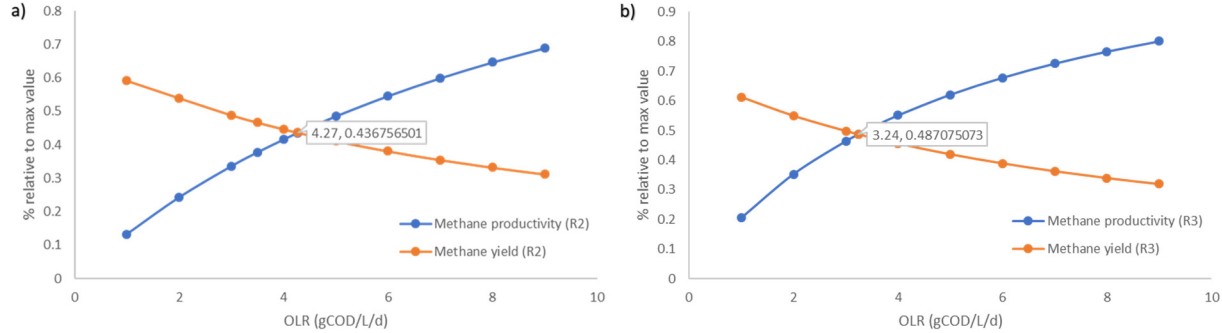

**Figure 8.** Graphical optimizations of the optimum MPR and MY of R2 and R3 using the OLR.

## 4. Discussion

The AD of the PM+CWM, PM+CWM+SWW, and PM+CWM+SBT substrate mixtures in a continuous operation mode using the three different EGSBs reactors yielded three main conclusions regarding the mixtures (see below).

1. The synergistic effects described by the batch model in [24] were also found in the continuous operation.
2. The maximum methane yields in the continuous operation of any mixture of these four substrates were predicted using the batch model and multiplying the $BMP_\infty$ by a coefficient between 0.7 and 0.8.
3. The employment of the Stover–Kincannon model showed that all three mixtures had a different kinetical behavior, which could even be noticed among the two triple mixtures.

The synergistic effects due to the addition of a third substrate were most likely related to the C/N values. The high C/N values in the three-substrate mixtures explained the good performances observed; however, a higher ratio was no indication of the better performance of mixture 3 versus 2. Hence, the performance of the co-digestion of these mixtures should not be oversimplified by the C/N values without having taken into account other influential factors. Nevertheless, the most recommended C/N values in the literature were from 0 to 30 [4,13,63]. We note that all our mixtures had a C/N value < 20 (Table 1), which strongly suggested the increased proportion of the carbon-rich-substrates within the mixtures.

The concept of integrating an EGSB reactor in a typical agricultural biogas plant is also of relevance. Compared to a typical agricultural biogas plant, where the representative HRT values are between 50 and 150 d [12,64], high-rate reactors provide an alternative system for the treatment of liquid substrates or their liquid fractions at much smaller HRTs. Substrates mixtures that have influenced HRTs should be applied as suggested by Paulose and Kaparaju [63]. They stated that a degradation rate follows an inverse function with the HRT depending on substrate complexity. Consequently, higher HRTs need to be applied and lower degradation rates were expected for lignin-rich substrates than for protein- or sugar-rich substrates. Agricultural biogas plants in Germany typically co-digest animal manure with either maize or grass silage [11]; therefore, higher HRTs are expected for the three-substrate mixtures digested in this paper due to their complexity. However, the differences in the HRTs were always noticeably large. Ruile et al. [12] studied 21 full-scale plants in the region of Baden-Württemberg (southern Germany), which performed either

single digestion or co-digestion of cattle manure, maize silage, and grass silage at different solid contents. They found that high values of degradability were reached at a HRT of $\geq$100 d. Thus, a more sophisticated scheme of treatment that involves multistage processes has been suggested for more efficient energy production [4,63]. Thus, the integration of a high-rate reactor in a typical treatment plant could lead to an increment in energy production, as found by Shen et al. [65] in their co-digestion of fruit/vegetable and food wastes in two stages (UASB+CSTR). This approach allowed them to work at higher OLRs and increase MPR values up to 15% over a single-stage digestion (UASB).

We were able to operate all three reactors for up to 1 d, where the tanks were refilled daily. Consequently, the daily preparation of the mixtures was a logistical and practical challenge. Also, the MYs obtained at a HRT of 1 d were the lowest among the three reactors. It was probably not ideal to run the reactors at such a low HRT; however, this was possible and can be especially useful when the demand for biogas is peaking or excess amounts of substrates need to be processed.

The results of the continuous operation were significantly influenced by the lack of proper mixing in the storage tanks and the seasonal behavior of the substrates. The latter was more obvious in the manure substrates and buffered by the addition of a third substrate. Mixture 2 was the least affected since it had the lowest ODM content in the feeds. Consequently, this mixture had the highest substrate homogeneity inside the reactor due to mixing by recirculation and increased biomass-substrate contact, which facilitated the operation [19,20].

The obtained results support the technical feasibility of the AcoD of liquid manure-based mixtures using EGSB reactors. Thus, it raises the possibility of designing new treatment concepts employing EGSB reactors for the AD of liquid agro-industrial mixtures while significantly reducing the required operating time.

The calculation of the hydraulic dimensionless numbers strongly suggested a CSRT behavior. The assumption of a single reactor with a CSTR behavior simplified the modeling of an EGSB reactor. This was strongly considered when applicable. Due to the lack of biomass sampling along the reactor, and without an adequate computer flow dynamics (CFD) model or RTD study, the consideration of one CSTR seemed the better option. However, the measurements of biomass concentration together with CFD modeling or an RTD study were highly recommended to thoroughly model a reactor [3,6].

Likewise, the operation intervals for an operation at a pilot plant scale were laid down for the two three-substrate mixtures, since both mixtures were likely more profitable than the two-substrate mixture considered. This was in the context of EEG. Furthermore, we recommend the development of more complex models which will allow for the simultaneous control of several process variables as well as describe the potential interactions involved within these variables.

## 5. Conclusions

The AcoD of the liquid fraction of PM+CWM and a third carbon-rich substrate such as SWW or SBT was successfully carried out in EGSB reactors, which were operated continuously. This work provides an alternative to typical CSTR systems used for manure and liquid manure treatments. The flow pattern of the studied reactors behaved similarly to a complete mixture reactor. Notably, the hydraulic behavior of our reactors was similar to those found in the literature. Moreover, the results from the batch test were successfully transferred to a continuous scale through the development of empirical and statistical modeling and the optimization of operating OLR intervals. We will consider more complex mechanistic models in the future. Further experiments are going to be carried out at the pilot plant scale in the Saerbeck bioenergy park using one automatically controlled 500 L EGSB reactor.

**Author Contributions:** Conceptualization, R.E.H.R. and J.H.; methodology, R.E.H.R. and J.H.; software, R.E.H.R. and J.H.; formal analysis, R.E.H.R.; investigation, R.E.H.R.; resources, D.B. and L.W.; data curation, R.E.H.R. and J.H.; writing—original draft preparation, R.E.H.R.; writing—review and

editing, J.H., J.T., D.B. and L.W.; visualization, R.E.H.R. and J.H.; supervision, J.T.; project administration, E.B. and D.B.; funding acquisition, E.B. All authors have read and agreed to the published version of the manuscript.

**Funding:** This research was funded under the Bio-Smart Project. The Bio-Smart project is funded by the Federal Ministry of Food and Agriculture based on a resolution of the German Bundestag (project management agency Fachagentur Nachwachsende Rohstoffe e.V. (FNR), FKZ: 22031318).

**Data Availability Statement:** The data presented in this study are available on request from the corresponding author.

**Conflicts of Interest:** The authors declare no conflict of interest.

## Abbreviations

| | |
|---|---|
| AD | Anaerobic digestion |
| HRT | Hydraulic retention time |
| AcoD | Anaerobic co-digestion |
| SRT | Solids retention times |
| UASB | Up-flow anaerobic sludge blanket |
| CSTR | Continuous stirred tank reactor |
| EGSB | Expanded granular sludge bed |
| PM | Piglet manure |
| CWM | Cow manure |
| SBT | Sugar beet |
| SWW | Starch wastewater |
| COD | Chemical oxygen demand |
| d | day |
| OLR | Organic loading rate |
| MPR | Methane productivity |
| MY | Methane yield |
| $\eta_{COD}$ | Removal efficiencies of chemical oxygen demand |
| $\eta_{BOD5}$ | Biological oxygen demand on the fifth day |
| PCA | Principal component analysis |
| $Pe_{axial}$ | Axial Peclet number |
| Re | Reynolds number |
| RMSE | Root-mean-square error |
| RTD | Residence time distribution |
| CFD | Computer flow dynamics |

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
