# Peer review of "Continuous Co-Digestion of Agro-Industrial Mixtures in Laboratory Scale Expanded Granular Sludge Bed Reactors"

_applsci, doi:10.3390/app12052295_

Round 1

Reviewer 1 Report

Manuscript ID applsci-1583614

Title

Continuous co-digestion of agro-industrial mixtures in laboratory scale expanded granular sludge bed reactors

The topic of the paper is likely to be of interest to many readers of Applied Sciences and worth to be considered for publication.

The article is feasible for publishing on the journal provided that some changes are made, as suggested by the comments below.

Page 2, line 59. Please better define what the Authors intend with “economic feasibility”. Moreover, there are some disadvantages of the anaerobic co-digestion, e.g. associated with the need of the simultaneous temporal and spatial presence, , of different substrates and in amounts that are compatible with the mixing ratio. Please add some consideration about it.

Page 2, line 88. The logical information flow of the introduction is not very logical! The Authors begin by discussing the benefit of the co-digestion, skip to the need to optimize HRT, even by decoupling it from SRT, and jump to high-rate systems. It is necessary to better explain the choice of combining codigestion with the type of reactor used.  

Page 3, line 109. Please replace “Mixture 1” instead of “Mixture number 1”.

Page 3, line 129. How was the pH regulated to the indicated value?

Page 8, line 262-273. No explanation was given for the failure observed with HRT 5d. MY is a more reliable parameter (being a specific index) than MPR (as also the Authors say at page 14, line 402). For this reason, I do not agree with the Authors concerning the fact that HRT = 5d is not a real “threshold” because MY is very low for lower HRT (i.e. 3d and 1 d). Other gas than methane was checked?

Page 11, line 314. The fluctuation of COD due to mixing absence do not explain the failure in terms of MY (that is specific at unit of volatile solids).

Page 11, line 328. I disagree: the performances of the 3 reactors are quite different. In particular, it seems that R2 and R3 work as duplicate whilst R1 had a failure for HRT = 5d. please give some explanation about this.

Author Response

Dear Sir/Madam,

All your points were addressed in the attached document. Have a nice day.

Kind regards,

Roberto Hernández

Reviewer 2 Report

The manuscript investigated the anaerobic digestion of three mixtures using panded granular sludge bed (EGSB) reactors. The results are presented in systematic manner. Quality of English is good. The manuscript may get acceptance after addressing following points:

  1. The novelty of study is needed to be highlighted clearly. Despite of presenting interesting results it seems like a report. The authors are advised to carefully revise introduction part. It must show why the research was conducted and it significance.
  2. A list of acronyms is required.
  3. Conclusion part can be extended by highlighting future directions in detail.

Author Response

(The authors gave the same response as above.)
